

# Metagenomic insights into surface water microbial communities of a South Asian mangrove ecosystem

Anwesha Ghosh[1], Ratul Saha[2] and Punyasloke Bhadury[1,3]

[1] Centre for Climate and Environmental Studies, Indian Institute of Science Education and Research Kolkata, Mohanpur, Nadia, West Bengal, India
[2] Wildlife and Habitats Division, WWF-India Sundarbans Landscape, Kolkata, West Bengal, India
[3] Integrative Taxonomy and Microbial Ecology Research Group, Department of Biological Sciences, Indian Institute of Science Education and Research Kolkata, Mohanpur, Nadia, West Bengal, India

## ABSTRACT

Estuaries are one of the most productive ecosystems and their productivity is maintained by resident microbial communities. Recent alterations driven by climate change have further escalated these stressors leading to the propagation of traits such as antibiotic resistance and heavy metal resistance in microbial communities. Surface water samples from eleven stations along the Thakuran and Matla estuaries of the Sundarbans Biosphere Reserve (SBR) of Sundarbans mangrove located in South Asia were sampled in monsoon (June) 2019 to elucidate resident microbial communities based on Nanopore sequencing. Metagenomic analyses revealed the widespread dominance of Proteobacteria across all the stations along with a high abundance of Firmicutes. Other phyla, including Euryarchaeota, Thaumarchaeota, Actinobacteria, Bacteroidetes and Cyanobacteria showed site-specific trends in abundance. Further taxonomic affiliations showed Gammaproteobacteria and Alphaproteobacteria to be dominant classes with high abundances of Bacilli in SBR_Stn58 and SBR_Stn113. Among the eukaryotic communities, the most abundant classes included Prasinophyceae, Saccharyomycetes and Sardariomycetes. Functional annotation showed metabolic activities such as carbohydrate, amino acid, nitrogen and phosphorus metabolisms to be uniformly distributed across all the studied stations. Pathways such as stress response, sulphur metabolism and motility-associated genes appeared in low abundances in SBR. Functional traits such as antibiotic resistance showed overwhelming dominance of genes involved in multidrug resistance along with widespread resistance towards commonly used antibiotics including Tetracycline, glycopeptide and aminoglycoside. Metal resistance genes including arsenic, nickel and copper were found in comparable abundances across the studied stations. The prevalence of ARG and MRG might indicate presence of pollutants and hint toward deteriorating ecosystem health status of Sundarbans mangrove.

Corresponding author
Punyasloke Bhadury,
pbhadury@iiserkol.ac.in

## INTRODUCTION

Estuarine habitats form transitional habitats with changing environmental conditions including dynamic salinity regimes, nutrient fluctuations and complex organic matter pools. In rapid urbanization, estuaries have been anthropogenically impacted by increasing human impacts leading to ecosystem degradation. Habitat loss, eutrophication, hypoxia, organic and inorganic pollutants, the spread of pathogens and ocean acidification are some of the direct consequences of rapidly changing global climatic issues (*Crain et al., 2009*). Unchecked degradation of estuarine habitats now poses risks to human wellbeing from increased bacterial pathogens, more instances of drug resistance among pathogens, harmful algal blooms and contaminated seafood (*Kite-Powell et al., 2008*; *Laws, Fleming & Stegeman, 2008*). It is thus imperative to investigate the microbial populations inhabiting estuaries to assess the magnitude of impacts on ecosystem health (*Port et al., 2012*).

Resident microbial communities play key roles in mediating ecosystem processes including nutrient cycling and thereby maintain productivity of ecosystems. Plankton are the main primary producers in the surface waters and support the productivity of ecosystems by fixing inorganic carbon (*Cloern, 1996*). Organic matter and energy are channelized through diverse assemblages of archaea, bacteria, ciliates, nano- and dinoflagellates, and amoebae, sometimes by viral lysis- and subsequently through small grazers to higher consumers (*Azam et al., 1983*; *Legendre & Rivkin, 2008*; *Strom, 2008*). Multiple combination of interactions and associations between microbes, known as the microbial network, aids in transfer of energy through the marine food web (*Trombetta et al., 2020*). Several studies have assessed the dynamicity of microbial food webs based on experimental and modelling approaches (*Fraust & Raes, 2012*; *Posch et al., 2015*; *Fuhrman, Cram & Needham, 2015*; *D'Alelio et al., 2016*). Associations and functioning of the microbial communities are strongly controlled by major stressors, such as surface water temperature and salinity variations which in turn also control nutrient dynamics in estuarine environments. Such interactions can become more complex by the presence of mangroves in the vicinity of estuaries.

Mangroves become sources of complex carbon including litterfall which contributes substantially to the dissolved organic matter pool in estuaries (*Sardessai, 1993*; *Dittmar et al., 2006*; *Kristensen et al., 2008*). As mangrove ecosystem become sink of organic matter from litterfall, it also accumulates nutrients from freshwater, seawater and anthropogenic wastewater. Fugong mangroves, located south of the Jiulong estuary is densely polluted by urban sewage (*Tian et al., 2008*) as are other mangroves such as Haikou and Sanya in China (*Li et al., 2019*). Nitrogen and phosphorus from upstream agricultural fields further enrich the dissolved nutrient pool in this region (*Liu, Peng & Li, 2012*). The influence of such stressors can only be inferred by extensive analyses of the structural and functional aspects of resident microbial communities. Metagenomics which includes sequencing of environmental DNA (eDNA) directly from environmental samples can be used to infer taxonomic and functional microbial diversity and community changes can be subsequently monitored over spatio-temporal scales in response to both

environmental and anthropogenic impacts (*Nogales et al., 2011*; *Port et al., 2012*). Metagenomics has provided key insights into microbial community structure and role of associated stressors in mangrove ecosystems such as in Malaysia, China and Brazil (*Priya et al., 2018*; *Cecoon et al., 2019*; *Li et al., 2019*; *Zhao et al., 2019*). Such stressors could alter microbial communities and disseminate traits such as antibiotic resistance. Even though the instances of development of such traits are on the rise, very little is known about factors responsible for rapidly increasing abundances of acquired pathogenesis in microbial communities especially in estuarine mangroves such as the Indian Sundarbans.

The Sundarbans lies in the delta of Ganga-Brahmaputra-Meghna (GBM) of South Asia and represents the largest contiguous mangrove forest in the world (*Gopal & Chauhan, 2006*). The mangroves span an area of ~10,000 km$^2$ across India and Bangladesh and home to seven broad estuaries that drain into the Bay of Bengal (*UNEP World Conservation Monitoring Centre, 2005*). Sundarbans is a UNESCO World Heritage Site and a RAMSAR site. The Sundarbans experience dynamic environmental conditions with rapidly changing nutrient concentrations in response to altering surface water temperature and salinity. Typical estuarine conditions rise from the mixing of freshwater flow from the rivers with the diurnal tides entering from the coastal Bay of Bengal (*Choudhury et al., 2015*). Salinity plays a crucial role in shaping the surface water microbial community assemblages of estuaries in Sundarbans (*Ghosh & Bhadury, 2018*, *2019*). The recently concluded Red List of Ecosystems indicated 'Endangered' status of the Indian Sundarbans based on assessment of diminishing fish populations while predicting further decline from reduced freshwater flow and other climatic events (*Sievers et al., 2020*). Metagenomic insights into the microbial communities of Sundarbans could thus be essential to gain crucial insights into the environmental dynamicity of the region.

The present study uses a metagenomic approach (a) to establish baseline information of microbial community structure from stations with comparable salinity values (b) to identify crucial genes that could provide key insights into the functioning of microbial communities of Sundarbans Biosphere Reserve of Sundarbans.

## MATERIALS AND METHODS

### Study site

The study was undertaken in the Indian Sundarbans Biosphere Reserve where 250 stations along Thakuran and Matla estuaries were sampled in monsoon of 2019. Out of these, 11 stations with comparable salinity values were selected for elucidation of bacterioplankton community structure (Fig. 1). Selected station numbers correspond to the original station designation where SBR_Stn2 is the 2nd of the 250 stations sampled. Upstream stations lie within close proximity to human inhabited areas and pristine regions of SBR. Downstream stations along the studied estuaries lie in the heavily protected reserve forest area and are within close proximity to the coastal Bay of Bengal. No human activities are permitted within the pristine regions. The downstream sampling stations are completely inaccessible to humans during monsoon season.
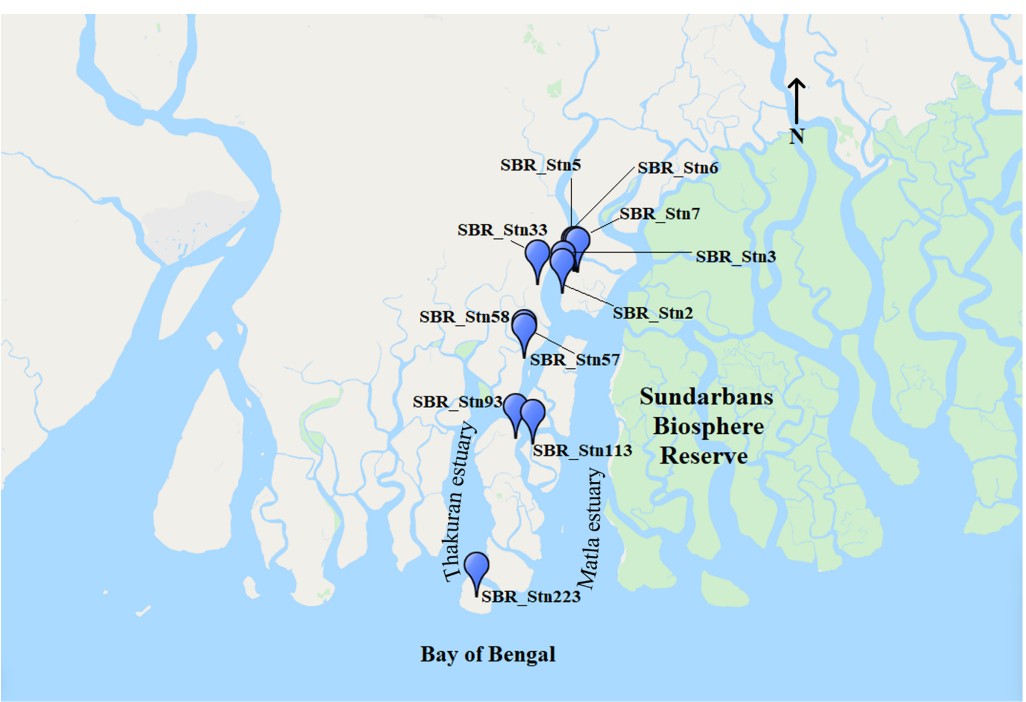

**Figure 1 Site map showing the Indian Sundarbans along with the position of Thakuran and Matla estuaries.** The blue points indicate the position of the sampling stations.

## Sampling

Sampling was conducted in two hundred and fifty stations selected along Matla and Thakuran estuaries in monsoon (June, 2019). Samples were collected on board *R/V Sundari*. Microbial communities were elucidated from 1 L of surface water samples collected from 11 stations following published protocols (*Ghosh & Bhadury, 2018*). Given salinity is a strong driving force that shapes microbial community structure in estuaries, stations exhibiting comparable salinity profiles would potentially harbour similar microbial communities and nullify variations resulting from sharp salinity differences. Environmental parameters including air temperature (AT; digital thermometer, Eurolab, Brussels, Belgium), surface water temperature (SWT; digital thermometer, Eurolab, Brussels, Belgium), salinity (Salt 6+ salinity probe; Eutech Instruments Pte Ltd., Ayer Rajah Crescent, Singapore) and pH (Eco testr pH2; Eutech Instruments Pte Ltd., Ayer Rajah Crescent, Singapore) were measured in triplicates at each station during the time of sampling. The handheld instrument probes used in this study were ATC enabled and calibrated in the laboratory and subsequently in the field with standards as per manufacturer's instructions before undertaking sampling each time during the study period.

## Nutrient analyses

Surface water samples were collected in triplicates for dissolved nutrient analysis. Dissolved nitrate (*Finch et al., 1998*), *ortho*-phosphate (*o*-phosphate) (*Strickland &*

*Parsons, 1972*), silicate and ammonium (*Liddicoat, Tibbitts & Butler, 1975*) concentrations were measured in triplicates using a UV-Vis spectrophotometer (U2900; Hitachi Corporation, Tokyo, Japan).

## Environmental DNA extraction

Microbial communities were concentrated by filtering one litre of water through a 0.22 μm 47 mm nitrocellulose filter paper (Pall, New York, NY, USA) using standard methodology (*Ghosh & Bhadury, 2018*) and the filters were processed immediately for environmental DNA (eDNA) extraction. The eDNA pool was extracted from each filter following published protocols (*Boström et al., 2004*): briefly, a sucrose salt lysis buffer (400 mM NaCl, 50 mM Tris-HCl, 20 mM EDTA, 750 mM sucrose, 10% SDS; Merck, India) was added to the filter paper and incubated for 30 min at 50 °C. Then, 5 μL Proteinase K (Amresco, Dallas, TX, USA) was added to the lysis buffer and subjected to incubation for 4 h at 55 °C. Subsequently, 10 μL Lysozyme (ThermoFisher Scientific, Dreieich, Germany) was added and the filters were incubated for 2 h at 37 °C. Phenol:Chloroform (Merck, India) was then added to the lysis buffer in the ratio of 1:1:2 and centrifuged at 16,000 rcf (radius 10 cm) for 12 min and the aqueous fraction was separated. The eDNA in the aqueous fraction was incubated overnight with 3M sodium acetate (Merck, India) and absolute ethanol (Merck, Darmstadt, Germany) was added for precipitation. The eDNA was pelleted at 16,000 rcf (radius 10 cm) for 12 min, air dried and dissolved in 30 μL 10 mM Tris-HCl (Merck, India). The extracted eDNA was quantified by agarose gel electrophoresis.

## Library preparation

Amplicon libraries were prepared using the Ligation sequencing kit (SQK-LSK109; Oxford Nanopore Technologies, Oxford, UK) and PCR barcoding kit (EXP-PCR096; Oxford Nanopore Technologies, Oxford, UK). Briefly, the protocol included end preparation, barcoding and sequencing adapter ligation. A total of 200 ng of purified extracted eDNA of each sample was end-repaired using NEBnext Ultra II End Repair Kit (New England Biolabs, Ipswich, MA, USA). This was cleaned using 1X AmPure beads (Beckman Coulter, Brea, CA, USA). Following this, barcode adapter ligation was performed with NEB blunt/ TA ligase (New England Biolabs, Ipswich, MA, USA) and again cleaned with 1X AmPure beads. The barcode adapter-ligated products were quantified using Qubit Fluorometer (Thermo Fisher Scientific, USA). Barcoded samples were cleaned up using 1.6X AmPure beads and pooled at equimolar concentrations. Pooled barcoded samples were end prepared using NEBNext Ultra II End Repair/dA-Tailing Module (New England Biolabs, Ipswich, MA, USA). End-repaired DNA was cleaned up with 1X AmPure beads.
The products were then adapter-ligated (AMX) using NEB blunt/TA ligase (New England Biolabs, Ipswich, MA, USA). The library mix was finally cleaned using AmPure beads and eluted in 15 μL elution buffer.

## Metagenomics data generation

Sequencing was performed on GridION X5 (Oxford Nanopore Technologies, Oxford, UK) using SpotON Flowcell R9.4 (FLO-MIN106) involving a 48-h sequencing protocol.

Nanopore raw reads (fast5 format) were base-called (fastq format) and demultiplexed using Guppy v2.3.4 (available from https://community.nanoporetech.com). Sequence data has been submitted to NCBI under the BioProject number PRJNA786468 and the data can be accessed publicly using the following link: https://trace.ncbi.nlm.nih.gov/Traces/study1/?acc=PRJNA786468&go=go.

## Data processing

Raw sequence files in fastq format were uploaded onto MG-RAST (*Meyer et al., 2008*). Sequence files were normalized following quality control. For quality control, artificial duplicate reads were removed, reads were trimmed based on quality and length. The MG-RAST pipeline was used for taxonomic identification and protein prediction using clustering and similarity-based annotation tools. Taxonomic annotation was performed against SILVA whereas proteins were annotated as SEED functions (*Overbeek et al., 2005*), KO terms, COG classes (*Tatusov et al., 2003*) and eggNOGs (*Jensen et al., 2016*). Extensive details about data processing steps can be read on the MG-RAST user manual (https://help.mg-rast.org/user_manual.html).

Identification and annotation of antibiotic-resistant genes (ARGs), mobile genetic elements (MGEs) and metal resistant genes (MRGs) were performed on the NanoARG platform in Fasta format (*Arango-Argoty et al., 2019*). NanoARG uses DeepARG/ARGMiner (*Arango-Argoty et al., 2018*) databases to detect ARGs in the database. For MGEs, the NanoARG-MGE database is used as a reference whereas MRGs are determined using BacMet.

# RESULTS

## *In-situ* environmental parameters

The AT and SWT ranged from 29.4–32.5 °C and 29.7–31.5 °C, respectively. Salinity ranged from 24.6 to 30.5 with salinity being lower in the downstream stations compared to upstream stations. The lowest salinity was recorded in SBR_Stn223. The pH did not show significant variation between the stations and ranged from 7.29–7.57. DO showed wide variation between the stations with values being lower in downstream stations (average 7 mg/L). DO was around 10 mg/L in SBR_Stn6 and SBR_Stn7. TDS and EC were comparable between the stations with the lowest values of TDS (9,320 ppm) and EC (18,400 µS/cm) recorded in SBR_Stn93. Secchi depth always showed wide variation between the stations with the highest value of 16.3 cm at SBR_Stn223 and the lowest value of 6 cm at SBR_Stn57. Measured values of environmental parameters have been summarized in Table S1.

## Dissolved nutrient concentrations

Dissolved nitrate ranged from 33–47.2 µM with the highest concentration at SBR_Stn33. All stations had a low concentration of dissolved *o*-phosphate (>1 µM) except SBR_Stn113 and SBR_Stn223. Dissolved ammonium was not detected in SBR_Stn93 whereas SBR_Stn57 had about 6 µM concentration. Most variation was observed in dissolved silicate concentrations. No dissolved silicate was detected in SBR_Stn6, Stn33 and Stn57.

High concentrations of dissolved silicate were detected in SBR_Stn93 (43.9 µM) and SBR_Stn2 (38.4 µM). The dissolved nutrient concentrations have been summarized in Table S1.

## Biological community description

Metagenomics reads generated from eDNA ranged from 374,471 to 668,680 reads (>500 bp) per sample. Detailed information on the total number of generated reads and the number of sequences affiliated to specific gene types are summarized in Table S2. The coverage number of all identified SSU rRNAs and functional genes were normalized for comparison of abundance across samples.

## Microbial community structure

Around 97–99% of sequence data were affiliated to bacterial taxa across the studied stations with the remaining sequences belonging to Eukaryota and Archaea. Only about 0.1–0.3% of the sequences showed affiliation with viruses. Taxonomic affiliation at the phylum level revealed the presence of seven abundant (>1% of total community) phyla across the studied stations (Fig. S1). Proteobacteria was found to be overwhelmingly abundant across studied stations with the lowest abundance observed in SBR_Stn58 and SBR_Stn113. The second most abundant phylum was Firmicutes with the highest abundance in SBR_Stn58 and SBR_Stn113. Apart from Proteobacteria and Firmicutes, there was a sizeable abundance of Bacteroidetes and Actinobacteria in the studied stations. The highest abundance of Actinobacteria was found in SBR_Stn5. Among Archaea, Euryarchaeota and Thaumarchaeota appear to be abundant across the studied stations of SBR.

At the class level, studied stations were dominated by Gammaproteobacteria (3–73% of the total community abundance) and Alphaproteobacteria (2–44% of the total community abundance). The highest abundance of Gammaproteobacteria was seen in SBR_Stn57 (~73%) and the highest abundance of Alphaproteobacteria was seen in SBR_Stn223 (~44%). An overwhelming abundance of Bacilli was seen in SBR_Stn58 (83%) and SBR_Stn113 (61%) (Fig. 2).

A total of 207 bacterioplankton families were detected from across the studied stations. Among these, the most abundant families included Vibrionaceae, Rhodobacteraceae, Pseudomonadaceae, Oceanospirillaceae, Moraxellaceae, Clostridiaceae, Campylobacteraceae and Bacillaceae (Fig. S2). The abundant bacterioplankton families did not show uniform distribution across all the studied sites. Bacillaceae showed the highest abundance in SBR_Stn58 (86% of total community) and SBR_Stn113(~68% of the total communities). Vibrionaceae showed high abundance in SBR_Stn5(24%), SBR_Stn7(32%), SBR_Stn33(18.6%) and SBR_Stn57(22.8%). Clostridiaceae showed the highest abundance in SBR_Stn5(38.8%) whereas Camphylobacteraceae was high in SBR_Stn7 and SBR_Stn93 (15.2 and 21.7% respectively). Other dominant bacterioplankton families including Rhizobiaceae, Burkholderiaceae, Bacteroidaceae and Alteromonadaceae showed low abundance (1–10%) and were uniformly distributed across the studied stations (Fig. S2).

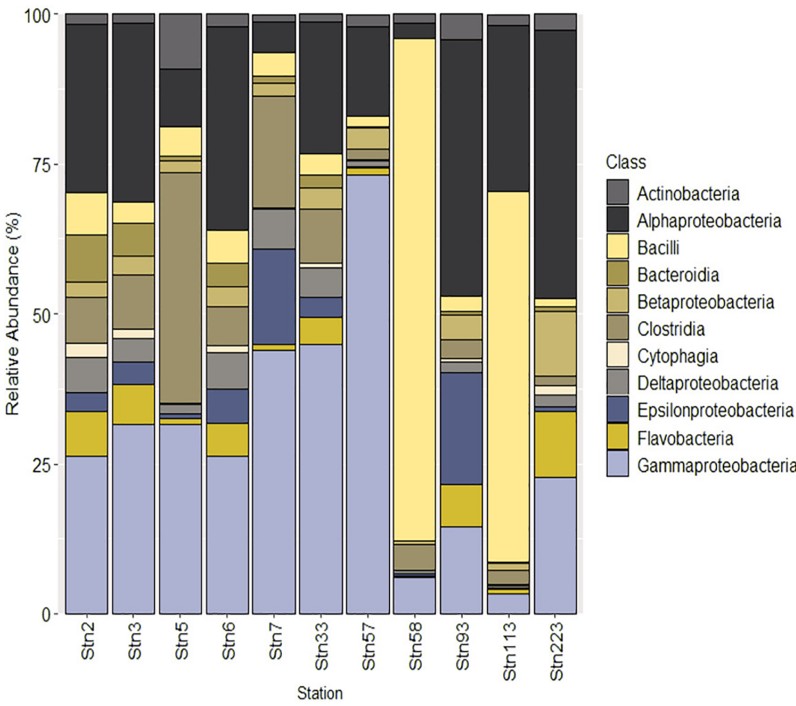

**Figure 2** Relative abundance of classes (domain: Prokaryota) identified from the studied stations of SBR during monsoon of 2019.

## Eukaryotic community structure

Only 0.2–1.3% of the sequences could be affiliated to the domain Eukaryota with the highest abundance in SBR_Stn223 (1.3%) and lowest in SBR_Stn58 (0.2%). Further taxonomic affiliation showed several fungal classes including Saccharomycetes, Schizosaccharomycetes, Sordariomycetes, Dothiodeomycetes, Eurotiomycetes, Leotiomycetes, Agariocomycetes, Tremellomycetes and Ustilaginomycetes (Fig. S3). All studied stations had comparable abundance of the fungal classes with the highest abundance of Saccharomycetes. Among diatoms, two classes including Bacillariophyceae and Coscinodiscophyceae showed widespread distribution across the studied stations. Bacillariophyceae had nearly equal abundances in all studied stations of SBR but Coscinodiscophyceae showed high abundance in SBR_Stn223 (Fig. S3). At the genus level, *Phaeodactylum* and *Thalassiosira* appeared in all the studied stations and *Odontella* showed high abundance only in SBR_Stn223.

## Functional level annotation

The microbial communities were subjected to functional level annotation to estimate functional traits and their distribution across the studied stations. Primary metabolic pathways including amino acid, carbohydrate, fatty acids, lipids and isoprenoids metabolisms were uniformly distributed by the microbial communities across the studied sites. The number of genes involved in fatty acids, lipids and isoprenoids was lower in

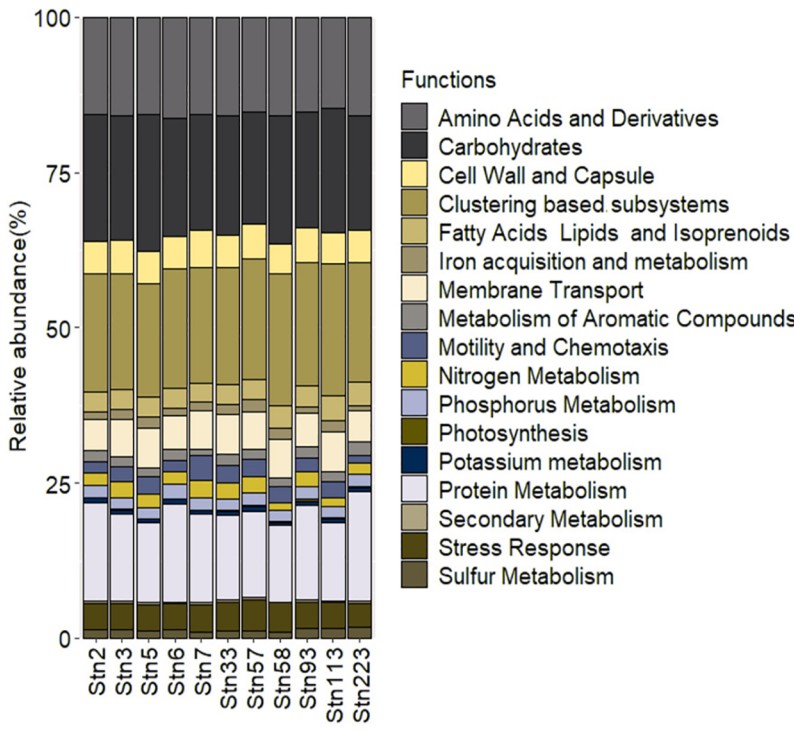

**Figure 3** Relative abundance of major functional pathways identified from the studied stations of SBR during monsoon of 2019.

abundance compared to amino acid and carbohydrate metabolisms (Fig. 3). Genes involved in the metabolism of nitrogen, potassium, phosphorus and sulphur were also widely distributed in microbial communities of SBR. A substantial number of genes were found which may be involved in the acquisition of iron from the environment. The presence of genes involved in the metabolism of aromatic compounds and stress response were found to be low in abundance but present in all the studied stations.

The identified functional signatures were further subcategorized to understand the ecological traits of the microbial communities of SBR. Iron acquisition capacities appeared to be mediated by siderophore formation which has lower abundance in SBR_Stn93 and SBR_Stn223 than the other stations (Fig. 4). The abundance of genes involved in anaerobic degradation of aromatic compounds and metabolism of central aromatic compounds did not show significant variations across the studied stations though their abundance in SBR_Stn2 and SBR_Stn7 was marginally lower than that of the other stations. Genes involved in pathways of bacterial cytostatics, differential factor and antibiotic generation, and biosynthesis of phenylpropanoids appeared in low abundance across the studied stations. The microbial communities appear to possess several genes involved in various stress response pathways including acid stress, cold and heat shock, desiccation stress and osmotic stress. Genes involved in periplasmic stress responses are low in abundance except in SBR_Stn7 and SBR_Stn57 (Fig. 4). The abundance of genes involved in inorganic and organic sulfur assimilations show uniform distribution across the studied stations.
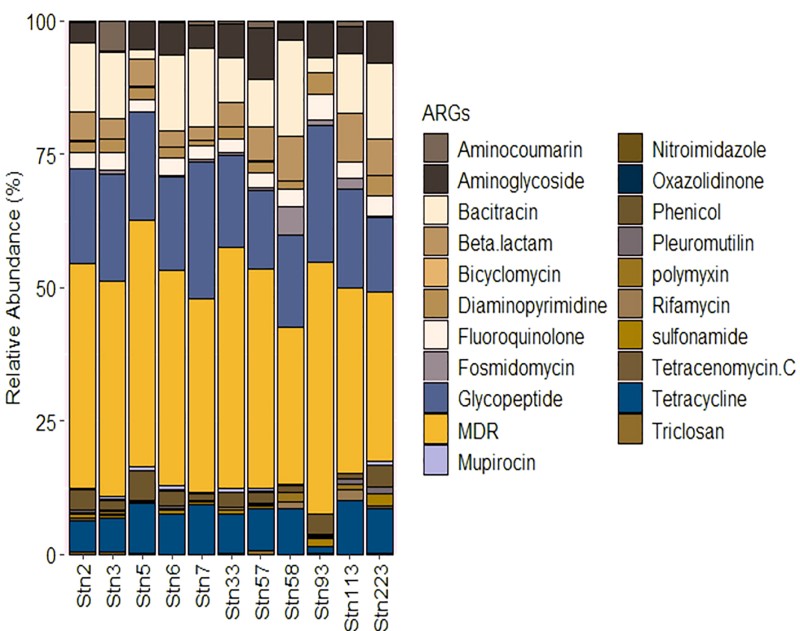

**Figure 4** Relative abundance of ARGs identified from the studied stations of SBR during monsoon of 2019.

## Antibiotic resistance genes (ARGs)

Functional analysis indicated the presence of a wide variety of genes possibly involved in conferring resistance to known antibiotics. Genes involved in possible multidrug resistance (MDR) showed highest abundance with ubiquitous distribution across the studied stations of SBR. The abundance of MDR genes did not show significant variation across the stations but was marginally higher in SBR_Stn93 (~47% of total ARGs) (Fig. 4). The genes found to be involved in coding for components of the MDR system were present in multiple copies across multiple studied stations and these were: *abe*MS (MATE family efflux transporter), *acr*ABF (Efflux pump), *ade*BCHKRS (Efflux pump membrane transporter), *bpe*EF (Efflux transporter periplasmic adapter), *cde*A (MATE family efflux transporter), *cme*B (Efflux pump membrane transporter), *emr*ABDEK (Export protein), *hmr*M (Resistance protein), *hp*1181 (Efflux transporter), *mar*AR, *mdt*BCDEGHKLNP (Resistance protein), *mep*A (Penicillin-insensitive murein endopeptidase), *mex*ABCDEFGHIJKLQRTVWX (Resistance protein), *opr*MNZ (Efflux system), *qac*G (Efflux protein), *sme*CDEFRS (Efflux transporter outer membrane subunit) and *tol*C (Efflux pump subunit).

Other genes involved in resistance to antibiotics including Tetracycline (inhibits protein synthesis by binding to bacterial 30S ribosomal subunit) and Glycopeptide (inhibits bacterial cell wall formation) showed an average abundance of ~7% and ~18% respectively and ubiquitously distributed across the stations (Fig. 4). The genes involved in Tetracycline resistance include *tet*ABDJLMNPQRTW, *otr*ABC, *tet*34, *tet*35, *tet*36, *tet*39 and *tet*41. Other genes including *tet*AB(46), *tet*AB(48), *tet*AB(60), *tet*B(46) and *tet*(42) were also identified. Resistance to glycopeptides may be owing to the presence of

*van*ABCDEGHBKILMNRSTUWXYZ along with *van*HB, *van*KI, *van*RI, *van*SD, *van*TG, *van*TrL and *van*YG1.

Station specific distribution of ARGs were found for Bicyclomycin, Mupirocin, Nitroimidazole, Oxazolidinone, Polymyxin, Tetracenomycin C and Triclosan with low abundance (<1%).

## Mobile genetic elements (MGEs)

Two MGEs including transposon and integron were identified in the studied stations. The abundance of integron was <0.1% and was not found in SBR_Stn2, SBR_Stn113 and SBR_Stn223. The abundance of transposon was also low in the studied stations with the highest abundance in SBR_Stn5. Associated MGEs including integrase, recombinase and transposase were widely distributed in all the studied stations with highest abundance of transposase. Two other enzymes namely, integrase and recombinase were also identified from all stations with the lowest abundance in SBR_Stn113 (Fig. S5).

## Metal resistant genes (MRGs)

Genes conferring resistance to several metals including nickel, molybdenum, cobalt and arsenic were identified from all the studied stations (Fig. S6). Nickel and arsenic specific MRGs were highest in abundance. Mercury resistance genes were found in all stations except SBR_Stn223. Zinc resistance genes were not identified from SBR_Stn5, SBR_Stn7 and SBR_Stn223. Gold resistance genes were low (>1%) in all stations except for SBR_Stn223 (6.9%) whereas tungsten resistant genes (0.9%) were found in the lowest abundance in SBR_Stn223.

## DISCUSSION

The Sundarbans mangrove ecoregion sits on the GBM delta of South Asia and forms innumerable rivulets and creeks along with seven major estuaries. Mixing of freshwater from the rivers with the saline water from coastal Bay of Bengal gives rise to typical estuarine conditions within the estuaries themselves. Detected environmental parameters including AT, SWT and salinity indicated typical tropical estuarine conditions. Large variations in DO concentrations were observed among the stations with higher DO values occurring in the more enclosed and low depth ones. In the upstream stations, high DO appears to be corresponding with high Secchi depth which could indicate higher phytoplankton abundance at these stations owing to more light penetration into the surface water. Higher concentrations of dissolved ammonium and nitrate appear to be typical of SBR as also seen previously in this region including parts of the Northern Indian Ocean (*Ghosh & Bhadury, 2019*). The low silicate concentrations could be due to heavy local precipitation during the time of sampling but further conclusions about such observations can only be drawn based on in-depth investigations across these stations encompassing seasonal scales.

In addition to the widely prevailing estuarine conditions, numerous micro-niches are formed within the narrow creeks and rivulets that connect estuaries and passes through large forested regions. Owing to the intricate pattern of dense mangrove forests, many

areas are severely light attenuated which impairs light penetration into the water column and influences phytoplankton community composition. Light penetration is further controlled by the suspended particulate matter load and depth of the studied stations which could hint towards observed low phytoplankton abundance in these stations. Phytoplankton abundance ranged from 5,000/L in the upstream stations to 20,000/L in the downstream stations. Previous data on chromophytic phytoplankton indicated the presence of Bacillariophyceae, Cryptophyceae and Haptophyceae with the ubiquitous presence of genera such as *Thalassiosira*, *Chaetoceros* and *Phaeocystis* (*Samanta & Bhadury, 2014*, *2016*). In an inter-annual monitoring study, it was been shown that phytoplankton abundance exhibited seasonal trends in their abundance between 1990 and 2007 with overwhelming dominance of *Coscinodiscus* throughout the study period (*De et al., 1991*; *Biswas et al., 2004*; *Biswas et al., 2010*). Low phytoplankton abundance of ~1,000–5,000/L with the dominance of *Coscinodiscus* and *Navicula* has also been reported from other regions of Sundarbans located in Bangladesh (*Hossain & Chowdhury, 2008*). Intertidal creeks including Kailash Khal in SBR have been shown to harbour low phytoplankton abundance (~4,000/L in pre-monsoon and post-monsoon, and ~2,000 in monsoon seasons) with the dominance of members of Bacillariophyceae (*Gogoi et al., 2019*). In the present study, *Thalassiosira* was found in all the studied stations which could be owing to the wide range of adaptive capabilities of this genus. Previous studies have shown metabolic capabilities of *Thalassiosira* to utilize a wide range of compounds including urea as a source of nitrogen (*Dortch, 1990*; *Bender et al., 2014*); specialized uptake proteins to enhance iron uptake for survival (*Kustka, Allen & Morrel, 2007*) and adjust rapidly to changing carbonate stoichiometry (*Bromke et al., 2013*). Variations in carbon dynamics would also alter processes including methanogenesis which could, in turn, control the structure of archaeal populations in Sundarbans. Members of Archaea including methanomicrobia and methanobacteria have been previously reported from anoxic sediment of Sundarbans and are known to be involved in methanogenesis (*Bhattacharyya et al., 2015*). Changes in dissolved oxygen concentrations might lead to a similar phenomenon and alter the Archaeal populations of the surface water from the studied sites. Anthropogenic pressures have also been shown to alter Archaeal communities where Euryarchaeal communities were found to vary in abundance in response. Adaptation of Archaeal communities was further demonstrated by their abilities to degrade compounds including benzoic acid, naphthalene and o-phthalate (*Mukherji et al., 2021*). The functional level annotation of surface water communities could also provide crucial information to understand anthropogenic pressures on the Sundarbans mangrove ecosystem.

Apart from diatoms, eukaryotic communities of the SBR stations were dominated by members of fungi. They contribute to particulate organic matter degradation in mangrove ecosystems and therefore are regarded as essential actors of the mangrove detrital food web (*Hyde & Lee, 1995*). Fungi associated with the phyllosphere are known to degrade mangrove litter (*Osono, 2006*). It has been estimated that 625 mangrove fungal species are belonging to phyla such as Basidiomycota, Ascomycota, Glomeromycota, Microsporia, among others (*Simões et al., 2013*; *Alsheikh-Hussain, Atenaiji & Yousef, 2014*). Some

studies describe the site-specific variations in fungal community structures; however very little is known about the fungal communities inhabiting the surface water of Sundarbans. Members of Saccharomycetes has been previously isolated from Sundarbans and shown to possess antimicrobial activities (*Simlai & Roy, 2012*). The predominance of Saccharomycetes in the studies stations of Sundarbans hinted towards unknown functions mediated by microbial communities. Therefore, in-depth exploration using such metagenomic surveys could reveal a wealth of information to deduce the functional significance of fungal communities inhabiting the surface waters of mangrove ecosystems of South Asia and beyond.

The bacterioplankton communities are overwhelmingly dominated by Proteobacteria and this group exhibited distinct spatial level separation among studied stations. The widespread presence of Proteobacteria across estuaries of Sundarbans have been previously documented (*Ghosh & Bhadury, 2018*, *2019*; *Mukherjee et al., 2020*) as well as reported from other mangrove ecosystems located in Brazil and Australia (*Nogueira et al., 2015*; *Shiau & Chiu, 2020*). Cyanobacteria appeared in low abundance indicating that they may be not important players in carbon cycling in Sundarbans throughout the year except in a particular season as observed earlier (*Bhadury & Singh, 2020*). The surface water of Sundarbans also harbours high abundance of Firmicutes typically reported in high numbers in mangrove sediment (*Nogueira et al., 2015*; *Halder & Nazareth, 2019*; *Fernández-Cadena et al., 2020*). The high abundance of Firmicutes has been attributed to the presence of sulphur oxidizing bacteria and owing to the pollution status of ecosystems (*Behara et al., 2014*; *Varon-Lopez et al., 2014*). Many members of Firmicutes are capable of spore formation and can withstand unfavourable environmental conditions (*Parkes & Sass, 2009*). In SBR, genes associated with spore formation were found to be higher in two stations (SBR_Stn58 and SBR_Stn113) which also had a high abundance of Firmicutes. A high abundance of genes associated with quorum sensing would also be owing to high Firmicutes abundance in SBR. Classes of Firmicutes including Clostridia and Bacilli were present in all stations. The highest abundance of Bacilli was seen in SBR_Stn58 and SBR_Stn113 whereas Clostridia were high in SBR_Stn5 and SBR_Stn7. The high abundance of Bacilli in these stations could be signatures of human interferences as members of this class including *Staphylococcus*, *Streptococcus* and *Enterococcus* often cause infections in humans (*Patterson, 1996*) as they harbour pathogenic genes. Virulence genes identified from these two stations have the potential to exhibit resistance to antibiotics such as methicillin, vancomycin, among others as also confirmed by high abundance of ARGs including MDR. Genes including *las*I, *lux*I and *agr*C have been reported to control quorum-sensing mediated biofilm development in marine coastal waters (*Rampadarath et al., 2017*). Gene that code for bacterial regulatory protein (LuxR) and known to be involved in quorum sensing was identified from all the studied stations of SBR. Prevalence of ARGs and quorum sensing genes could indicate the possibility of novel pathogenic microbes or opportunistic pathogens inhabiting SBR.

Class level affiliation showed high abundance of Gammaproteobacteria and Alphaproteobacteria in the studied stations of SBR and have been also reported from other coastal ecosystems globally (*Sogin et al., 2006*; *Danovaro et al., 2010*; *Williams et al., 2010*;

*Zinger et al., 2011*; *Lyra et al., 2013*; *Sun et al., 2013*). High Alphaproteobacterial abundance was found in low salinity stations of SBR and such a trend has been also observed in the Pearl estuary which is home to mangroves (*Liu et al., 2015*). Alpha- and Gammaproteobacteria commonly show succession under dynamic saline conditions (*Wu et al., 2006*). Alphaproteobacteria showed a positive but weak correlation with salinity in estuaries such as the Delaware (*Benlloch et al., 2002*; *Kirchman et al., 2005*) but in the SBR stations, it showed a strong negative correlation with salinity. Members of Alphaproteobacteria are metabolically diverse and can sustain in a wide range of ecosystems as they possess an intricate group of general stress response (GSR) genes. These genes can help towards combating stresses including heat shock and low pH (*Fiebig et al., 2016*). Though the abundance of stress response genes, especially those of osmotic stress, could not be directly correlated with salinity variation across the studied stations, the abundance of extracytoplasmic function (ECF) σfactor family gene, *ecf*G, an internal component of GSR in Alphaproteobacteria, hinted towards adaptive capabilities of dominant bacterioplankton in SBR.

Family level taxonomic affiliation indicated the presence of Vibrionaceae and Rhodobacteraceae to be more widely distributed in stations of SBR. Members of Rhodobacteraceae are planktonic and functionally involved in cycling of sulphur and carbon (*Pujalte et al., 2014*). There is high abundance and dominance of Rhodobacteraceae in estuarine environments including in Sundarbans (*Rappé, Kemp & Giovannoni, 1997*; *Crump, Armbrust & Baross, 1999*; *Ghosh & Bhadury, 2019*). Similarly, being a marine taxon, members of Vibrionaceae are commonly encountered in coastal ecosystems and are of special importance concerning human health (*Vezzulli et al., 2016*). Vibrionaceae including the genus *Vibrio* is an estuarine heterotrophic taxon that is frequently found in close association with plants and animals (*Lee & Ruby, 1994*; *Nyholm & Nishiguchi, 2008*; *Geng et al., 2014*). The presence of Vibrionaceae is in high abundance, especially near human settlements, such as seen in SBR_Stn 5, 7, 33 and 57 are of concern as the copiotrophic nature of these bacteria enable them to rapidly increase in abundance and can lead to pathogenicity. Previous data indicates high abundance of human pathogens such as Legionellales in sediment samples from Sundarbans but this family was found in low abundance in the surface water possibly hinting towards fine-scale variations in community structure between sediment and surface water ecosystems (*Basak et al., 2014*).

The presence of high abundance of potentially pathogenic bacteria would provide key insights into the prevalence of ARGs across studied stations of SBR (Sundarbans). Antibiotics commonly used in animal farming and human health sectors include sulfonamides, fluoroquinolones and aminoglycosides and ARGs against such antibiotics have been previously reported from contaminated ecosystems like the South China Sea and Pearl estuary (*Chen et al., 2013*). The abundance of such ARGs was low in the studied stations possibly owing to the pristine nature of SBR. But the high prevalence of MDR genes poses a potential threat to the emergence of new pathogenic bacteria from this region. The distribution of MDR across mangrove ecosystems including in Saudi Arabia, Brazil and India appears to be uniform with increasing abundance over time scales
(*Imchen et al., 2018*). Genes such as *van*B-01 and *van*HB could give rise to vancomycin resistance enterococci (*Zhao et al., 2020*). Additionally, the presence of MGEs such as integron-integrase gene could result in the horizontal transfer of ARGs but the low abundance of these genes in SBR could be owing to limited human interference in this region. Even after the death of bacterial cells, ARGs could sustain for long periods in estuarine environments by attachment to sediment particles. Such ARGs, termed as eARGS, can adsorb to particles or organic matter and accumulate in sediments (*Pietramellara et al., 2009*). The high SPM load in the stations of SBR could hence become a medium for ARG transmission. MGEs have been found to proliferate ARGs in estuaries such as the Haihe in China and increases the possibility of pathogens becoming MDR (*Zhao et al., 2020*). Tracking the changes of ARGs and MGEs at a seasonal scale would hence become important in pristine ecosystems such as the Sundarbans and across South Asia.

The presence of ARGs in an environment is also linked to the presence of metals (*Knapp et al., 2017*; *Zeng et al., 2019*). Mangroves being vulnerable to pollutants including heavy metals could hence harbour a high abundance of ARGs (*Imchen & Kumavath, 2020*). Analysis of samples from Sundarbans has also shown a strong correlation between ARGs and heavy metals (*Bhattacharyya et al., 2019*). Several studies have explored heavy metal pollution in mangrove ecosystems and the capability of mangroves to retain these hazardous pollutants (*Fernandes & Nayak, 2012*; *Bodin et al., 2013*; *Li et al., 2016*). Metals such as lead, manganese, copper, zinc, iron and cadmium are retained by mangrove roots (*Chiu et al., 1995*; *Tam & Wong, 2000*). Bacterial communities are strongly influenced by metal concentrations in mangroves such as Beihai, Fangchenggang, Hainan, Hong Kong, Shenzhen, Yunxiao and Zhanjiang in South China (*Meng et al., 2021*). The abundance of MGEs and ARGs in the studied stations hint towards a potential increase in pollution levels in the Sundarbans. The high abundance of ARGs has also been reported from sediments of Sundarbans. Widespread distribution of the $bla_{TEM}$ gene indicated anthropogenic influences in Sundarbans. Additionally, resistance to commonly used antibiotics including ampicillin, kanamycin, vancomycin and tetracycline has also been reported from sediment samples of Sundarbans (*Bhattacharyya et al., 2019*).

## CONCLUSIONS

Mangroves play crucial role in environmental protection by the retention of harmful pollutants but increased anthropogenic pressures are making them highly sensitive to disturbances. In-depth analysis of structural and functional microbial communities could hence give us crucial insights about the ecosystem level functioning of mangroves and help predict possible environmental degradation owing to rapidly changing anthropogenic activities around mangrove ecoregions including the Sundarbans mangrove of South Asia. Such information is critical for national park managers to effectively intervene and manage vulnerable yet biodiversity rich ecosystems such as the Sundarbans in South Asia. This is the first study of surface water microbial communities undertaken in a South Asian mangrove based on Nanopore sequencing. The much-needed baseline information on the metagenome of microbial communities in Sundarbans can also help towards

understanding biogeographic patterns, prevalence and functions of key microbial communities in regional seas and oceans, understanding the effects of multiple stressors such as pollutants or changing pH (ocean acidification scenario) on functions of surface water microbial communities including from the Northern Indian Ocean and their regional significance that may ultimately lead to an improved estimation of carbon cycling in the coastal waters of South Asia.

## ACKNOWLEDGEMENTS
The authors thank the boat crew of R/V Sundari for their assistance during sampling.

### Funding
This work is supported by SwarnaJayanti Fellowship of Department of Science & Technology, Government of India (DST/SJF/E&ASA-01/2017-18) and WWF-India Collaborative Grant awarded to Punyasloke Bhadury. The funders had no role in study design, data collection and analysis, decision to publish, or preparation of the manuscript.

### Grant Disclosures
The following grant information was disclosed by the authors:
Department of Science & Technology, Govt of India: DST/SJF/E&ASA-01/2017-18.
WWF-India Collaborative Grant awarded to Punyasloke Bhadury.

### Competing Interests
Punyasloke Bhadury is an Academic Editor for PeerJ.

### Author Contributions
- Anwesha Ghosh conceived and designed the experiments, performed the experiments, analyzed the data, prepared figures and/or tables, authored or reviewed drafts of the paper, and approved the final draft.
- Ratul Saha analyzed the data, authored or reviewed drafts of the paper, and approved the final draft.
- Punyasloke Bhadury conceived and designed the experiments, performed the experiments, analyzed the data, prepared figures and/or tables, authored or reviewed drafts of the paper, and approved the final draft.

### Data Availability
The raw sequence reads are available at NCBI: PRJNA786468.

### Supplemental Information
Supplemental information for this article can be found online at http://dx.doi.org/10.7717/peerj.13169#supplemental-information.

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
