# Peer review of "Metagenomic insights into surface water microbial communities of a South Asian mangrove ecosystem"

_PeerJ, doi:10.7717/peerj.13169_

## Round 0.1 · original submission · Major Revisions

The reviews are generally favorable but highlight a number of issues that should be addressed in a revision. Please respond to all reviewer comments.

Reviewer 1 ·

Basic reporting

The manuscript titled “Metagenomic insights into surface water microbial communities of a South Asian mangrove ecosystem” describes the structure and function of microbial communities in the surface water of selected sampling stations with comparable salinity in Sundarbans mangrove estuary situated in the north-eastern coast of India. The authors took a metagenomic approach and used nanopore sequencing chemistry to decipher the microbial structure and function in the above-mentioned samples. The detailed metagenomic analyses revealed the widespread dominance of proteobacteria across all the stations along with a high abundance of Firmicutes. Other phyla, including Euryarchaeota, Thaumarchaeota, Actinobacteria, Bacteroidetes, and Cyanobacteria showed site-specific abundance. Moreover, Prasinophyceae, Saccharyomycetes, and Sardariomycetes were found to be the abundant eukaryotic class in the studied samples. Functional annotation analyses revealed the identification of metabolic pathways for carbohydrate, amino acid, nitrogen, and phosphorus metabolism across all the studied stations. Furthermore, antibiotic resistance and heavy metal resistance/mobilization pathways were also identified within the dataset. Together the results presented in the manuscript are important and can be explored further through extensive laboratory experiments. Unfortunately, authors completely ignored published and available pieces of literature from the same mangrove wetland estuary and this makes the present study considerably weak. They have only cited their own research works and have not even considered discussing other published research articles on mangrove microbial community analysis in India (Sundarbans, Goa, Kerala, etc.). A comparative analysis with published and available microbial community diversity analysis from this and other mangrove wetland systems might help authors to strengthen their results. The manuscript in its present form is not acceptable and warrants further comparative analysis with existing pieces of literature.

Experimental design

1. Please describe the logic behind the selection of sampling stations based on comparable salinity. In Sundarbans, salinity has a temporal behavior and therefore raises a question of why authors didn’t look into the temporal variability of microbial communities in these sampling stations?
2. Line 117: “pristine region”--- what does this mean? Did the authors use any marker to select this as a pristine environment?
3. Line 119-120: “No human activities are permitted within the pristine regions.”---
Please define pristine area? Moreover, as per the site selection, they are not in the core area, where human activity is restricted. Also, many of the sampled locations are often used for tourist activities. Please clarify and amend accordingly.
4. Line 125: ’11 stations’--- I couldn’t find any mention of biological or technical replicates? Did the authors consider the collection, processing, and sequencing replicates???
5. Line 114: When in pre-monsoon 2019? Please mention the month as the pre-monsoon extends for a few months!!!

Validity of the findings

Major comments:
1. Figure 1: The map of the sampling sites lacks proper annotations and wrongly represents the name of estuaries. What is shown here as the Thakuran estuary is actually the Matla and vice-versa. Also, the name of the rivers in the upper reaches is omitted, making it incoherent. The map scale is too low is unable to accurately determine the character of the sampling stations. Moreover, the coordinates of the 11 sampling sites are missing and it is unclear how the other sites among the 250 do not have comparable salinity levels. Also, there is a slight conflict in statements in lines 117-119 where the region is referred to be under 'heavily protected reserved forest' region yet states that they are inhabited and also geomorphologically this region is predominantly under the influence of tidal estuarine ecosystem rather than the coastal ecosystem. Even the statement of the entire region being inaccessible during the monsoons is largely exaggerated.
2. In the Abstract, the authors mention that besides Proteobacteria and Firmicutes, other phyla, including Euryarchaeota, Thaumarchaeota, Actinobacteria, Bacteroidetes, and Cyanobacteria showed site-specific abundance. However, this has not been extensively addressed further in the manuscript. For example, finding an abundance of Euryarchaeota in Sundarban mangrove has been reported previously and has been implicated for bioremediation [Bhattacharyya et al. (2015), Archaea (DOI: 10.1155/2015/968582); Mukherji et al. (2020), Ecotox. Env. Safety 195:110481]. Please discuss such possibilities in the present study.
3. Introduction (line 90-91): “Even though the instances…., very little is known about factors responsible for rapidly increasing abundances of acquired pathogenesis….the Indian Sundarbans.” --- Several articles are available on microbial community diversity and their correlation with the anthropogenic activities in the sediments of Sundarban (Mukherji et al, Ecotox. Env. safety 2020; Bhattacharyya et al. Sci Total Env 2019; Basak et al. Genomics Data 2016; Bhattacharyya et al. Archaea 2015; Basak et al. Genomics Data 2015; Basak et al. Microbiol Ecol 2015). It’s clear from the published literature and the data presented in the present manuscript that there exists considerable similarity in the microbial communities of the sediment and surface water of Sundarbans mangrove. Please discuss and compare published datasets while introducing your hypothesis. This will not dilute your contribution rather give strength to it.
4. Line 193-197: Unit of salinity is not clear from descriptions. In line 193, it seems salinity is presented in PSU (Practical Salinity Units). While in line 196, salinity is presented in mg/L. Please clarify and maintain uniformity throughout the manuscript.
5. Line 217-225: Please provide a quantitative comparative assessment (% reads etc.) to facilitate readers to understand variations comparatively.
6. Section ‘Microbial community structure’ (line 214-242): Although archaea are reported as significantly abundant in the microbial community, no detail has been provided about their spatial variation or abundance. Please provide details.
7. Line 270: Description of genes responsible for ‘anaerobic degradation is provided. However, no genes involved in aerobic degradation have been discussed. Didn’t the authors find any such metabolic genes within the dataset? As this study involves surface water, it is expected to identify genes involved in aerobic metabolism. In any case, please discuss this in detail.


Minor comments:
1. BioProject number PRJNA786468 is not accessible for review purposes.
2. Please check the language throughout and remove typos.

Additional comments

no comment

·

Basic reporting

Methods:

Sunderban has a number of estuaries, what is the specific reason for sampling from Thakuran and Matla estuaries?

The accession number of the samples in MG-RAST is missing. Raw reads submissions to NCBI are also missing.

As per the methods, samples were collected from 250 stations, however, the results indicate that there are only 11 datasets. Kindly provide how were the samples pooled or managed.

Results:

In the results section, functional annotation identified "cold and heat shock" resistance, what could be the reason?

Have you done Differently Abundant Genera/Gene (DAG) analysis? It seems to be missing. Or what is the reason for avoiding DAG analysis?

Discussion:

Several ARGs were identified in this study, was there any association between the abundance of those ARGs and the antibiotics used in the Sunderban (or nearby areas)?

Experimental design

The MS is interesting since it's one of the first metagenomic studies to do waste analysis from mangrove forests in India (Sunderban). However, there seems to be a lack of experimental design as no proper comparative analysis could be seen throughout the MS. Furthermore, statistical analysis is also lacking.

Validity of the findings

The beginning part of the abstract seems to convey the message that ARG and heavy metal resistance are enriched due to climate change. Although it could be true to some extent, the enrichment could be mainly due to pollution rather than climate change.

The reporting is simply a catalogue of the microbiome and the diversity of ARGs, it does not provide any information on the reasons or the patterns of occurrence. Hence, the abstract should be more informative and novelty is needed.

Additional comments

The MS may send for a major revision

·

Basic reporting

The article was well-written in clear and unambiguous, professional English throughout the main text. Nonetheless, a few scientific names were misspelled (please, verify).
The cited references are adequate, up-to-date, and cover the main theme of the article (Mangroves, Microbiome, and Metagenomics). The article is well-structured, the figures are clear, self-explained, and sufficient to graphically depict the main obtained results. Furthermore, Supplementary Table details the metagenomic results. Raw data is shared as a BioSample deposit in NICBI. The objectives are clear and are achieved and well-discussed.

Experimental design

This is an original primary research study and falls within the aims and scope of the PeerJ journal. The research question is well-defined, is original, and relevant, clearing stating how the study fills the posed knowledge gaps. The study is carried out using rigorous technical standards (e.g.: adequate sampling sufficiency, and replicates), the methodology is described in detail and permits the replicability of the study.

Validity of the findings

I consider this is an original study, clearly demonstrating the feasibility of the use of Nanopore Technology to shotgun metagenomics. The results are robust and statistically controlled, and the conclusions respond to the posed objectives in an adequately manner.

Additional comments

I would like that the authors clarify to me the following doubts:

1. Why did the community structure analysis was restricted to the more inclusive taxonomical categories (genera are not analyzed and discussed, for instance)? Is there any technical restriction? If yes, why?

2. How may this study be used by policymakers in order to avoid or mitigate more environmental problems (e.g.: pollutants) in this important area?

3. How do you compare the Nanopore technology with Illumina Technology for performing shotgun metagenomics. How the former can be ameliorated?

---

## Round 0.2 · Minor Revisions

Thank you for responding to the reviewers' comments. The manuscript is acceptable, but I cannot fully accept it for publication until the data is publicly available. Please activate your NCBI accession, and then I can accept the paper.

Reviewer 1 ·

Basic reporting

no comment

Experimental design

no comment

Validity of the findings

no comment

Additional comments

The authors have sufficiently addressed the reviewers' comments.

·

Basic reporting

Authors addressed all the comments which i have raised previous review

Experimental design

Improved

Validity of the findings

Improved

Additional comments

No comments

---

## Round 0.3 · accepted · Accept

Thanks for responding to all comments and for activating your NCBI accession.